# Targeted Adversarial Perturbations
# for Monocular Depth Prediction

**Alex Wong**
UCLA Vision Lab
alexw@cs.ucla.edu

**Safa Cicek**
UCLA Vision Lab
safacicek@ucla.edu

**Stefano Soatto**
UCLA Vision Lab
soatto@cs.ucla.edu

## Abstract

We study the effect of adversarial perturbations on the task of monocular depth prediction. Specifically, we explore the ability of small, imperceptible additive perturbations to selectively alter the perceived geometry of the scene. We show that such perturbations can not only globally re-scale the predicted distances from the camera, but also alter the prediction to match a different target scene. We also show that, when given semantic or instance information, perturbations can fool the network to alter the depth of specific categories or instances in the scene, and even remove them while preserving the rest of the scene. To understand the effect of targeted perturbations, we conduct experiments on state-of-the-art monocular depth prediction methods. Our experiments reveal vulnerabilities in monocular depth prediction networks, and shed light on the biases and context learned by them.

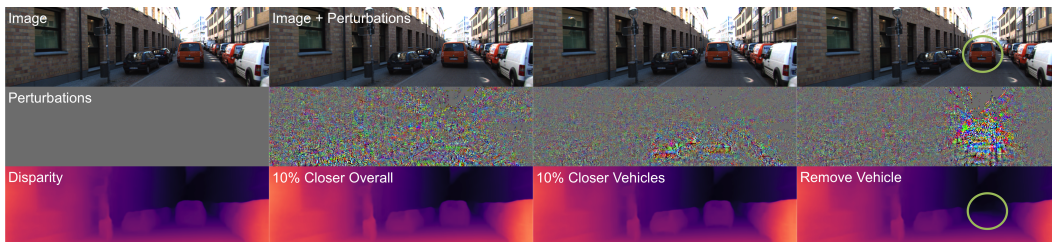

Figure 1: **Altering the predicted scene with adversarial perturbations.** Top to bottom: input image; adversarial perturbations with upper norm of $2 \times 10^{-2}$; predicted scene visualized as disparity. Left to right: original image and predicted scene; overall scene altered to be 10% closer; all vehicles altered to be 10% closer; vehicle in the center of the road is removed by perturbations.

## 1 Introduction

Consider the image shown in the top-left of Fig. 1, captured from a moving car. The corresponding depth of the scene, inferred by a deep neural network and visualized as disparity, is shown underneath. Can adding a small perturbation cause the perceived vehicle in front of us disappear? Indeed, this is shown on the rightmost panel of the same figure: The perturbed image, shown on the top-right, is indistinguishable from the original. Yet, the perturbation, amplified and shown in the center row, causes the depth map to be altered in a way that makes the car in front of us disappear.

Adversarial perturbations are small signals that, when added to images, are imperceptible yet can cause the output of a deep neural network to change catastrophically [37]. We know that they can fool a network to mistake a tree for a peacock [27]. But, as autonomous vehicles are increasingly employing learned perception modules, mistaking a stop sign for a speed limit [8] or causing obstacles to disappear is not just an interesting academic exercise. We explore the possibility that small perturbations can alter not just the class label associated to an image, but the inferred depth map, for instance to make the entire scene appear closer or farther, or portions of the scene, like specific objects, to become invisible or be perceived as being elsewhere in the scene.

When semantic segmentation is available, perturbations can target a specific category in the predicted scene. Some categories (e.g. traffic lights, humans) are harder to attack than others (e.g. roads, nature). When instance segmentation is available, perturbations can manipulate individual objects, for instance make a car disappear or move it to another location. We call these phenomena collectively as *stereopagnosia*, as the solid geometric analogue of prosopagnosia [4].

Stereopagnosia sheds light on the role of context in the representation of geometry with deep networks. When attacking a specific category or instance, while most of the perturbations are localized, some are distributed throughout the scene, far from the object of interest. Even when the target effect is localized (e.g., make a car disappear), the perturbations are non-local, indicating that the network exploits non-local context, which represents a vulnerability. Could one perturb regions in the image, for instance displaying billboards, thus making cars seemingly disappear?

We note that, although the adversarial perturbations we consider are not universal, that is, they are tailored to a specific scene and its corresponding image, they are somewhat robust. Blurring the image after applying the perturbations reduces, but does not eliminate, stereopagnosia. To understand generalizability of adversarial perturbations, we examine the transferability of the perturbations between monocular depth prediction models with different architectures and losses.

## 2  Related Work

Adversarial perturbations have been studied extensively for classification (Sec. 2.1). We focus on regression, where there exists some initial work. However, we study the *targeted* case where the entire scene, a particular object class, or even an instance is manipulated by the choice of perturbation.

### 2.1  Adversarial Perturbations

The early works [15, 37] showed the existence of small, imperceptible additive noises that can alter the predictions of deep learning based classification networks. Since then many more advanced attacks [27] have been proposed. [26] showed the existence of universal perturbations i.e. constant additive perturbations to degrade the accuracy over the entire dataset.

More recently, [30] studied transferability of attacks across datasets and models. [31] derived lower bounds on the magnitudes of perturbations. [29] studied the attacks in semi-supervised learning setting. [34, 38] proposed methods to enhance robustness to adversarial attacks. [20] extended adversarial attacks beyond small additive perturbations. [19] showed that the existence of adversarial attacks makes deep networks more predictive.

Despite the exponentially growing literature on adversarial attacks for the classification task, there only have been a few works extending analysis of adversarial perturbations to dense-pixel prediction tasks. [42] studied adversarial perturbations for detection and segmentation. [17] demonstrated targeted universal attacks for semantic segmentation. [40] studied non-targeted perturbations for stereo, and [35] used patch attacks for optical flow. [28] examined universal perturbations in a data-free setting for segmentation and depth prediction to alter predictions in arbitrary directions. Unlike them, we study *targeted* attacks where network is fooled to predict a specific target.

Our goal is to analyze the robustness of the monocular depth prediction networks to different targeted attacks to explore possible explanations of what is learned by these models. With a similar motivation, [5] inserted various objects into the images and [18] identified a small set of pixels from which a network can predict depth with small error. Unlike them, we analyze the monocular depth networks by studying their robustness against *targeted adversarial* attacks.

### 2.2  Monocular Depth Prediction

[6, 7, 21, 22, 23, 44] trained deep networks with ground-truth annotations to predict depth from a single image. However, high quality depth maps are often unavailable and, when available, are expensive to acquire. Hence, trends shifted to weaker supervision from crowd-sourced data [3], and ordinal relationships amongst depth measurements [10, 47].

Recently, supervisory trends shifted to unsupervised (self-supervised) learning, which relies on stereo-pairs or video sequences during training, and provides supervision in the form of image reconstruction. While depth from video-based methods is up to an *unknown* scale, stereo-based methods can predict depth in *metric* scale because the pose (baseline) between the cameras is known.

To learn depth from stereo-pairs, [11] predicted disparity by reconstructing one image from its stereo-counterpart. Monodepth [13] predicted both left and right disparities from a single image

---
**Algorithm 1** Proposed method to calculate targeted adversarial perturbations for a regression task.
---
**Parameters:** Learning rate $\eta$, noise upper norm $\xi$.
**Inputs:** Image $x$, target depth map $d^t(x)$, pretrained depth network $f_d$.
**Outputs:** Perturbation $v_N(x)$.
**Init:** $v_0(x) = 0$.
**for** $n = 0 : N - 1$ **do**
$\quad v_n(x) = \text{CLIP}\big(v_n(x), -\xi, \xi\big)$
$\quad$ Calculate $\ell(x, v_n(x), d^t(x), f_d)$ as defined in Eqn. 1.
$\quad v_{n+1}(x) = v_n(x) - \eta \nabla \ell(x, v_n(x), d^t(x), f_d)$
**end for**
$v_N(x) = \text{CLIP}\big(v_N(x), -\xi, \xi\big)$

---

and laid the foundation for [32, 33, 41]. To learn depth from videos, [25, 46] also jointly learned pose between temporally adjacent frames to enable image reconstruction by reprojection. [39, 43] leveraged visual odometry, [9] used gravity, [2, 24] considered motion segmentation, and [45] jointly learned depth, pose and optical flow. Monodepth2 [14] explored both stereo and video-based methods and proposed a reprojection loss to discard potential occlusions. PackNet [16] used 3D convolutions.

To study the effect of adversarial perturbations, we examine the robustness of unsupervised methods Monodepth [13], and the state-of-the-art Monodepth2 [14], PackNet [16] for outdoor scenes, and supervised method VNL [44] for indoors. While Monodepth2 proposed both stereo and video-based models, we choose their stereo model because the predicted depth is in *metric* scale, which enables us to study perturbations to alter the scale of the scene without changing its topology.

In Sec. 3, we discuss our method. We show perturbations for altering entire predictions to a target scene in Sec. 4 and localized attacks on specific categories and object instances in Sec. 5. We discuss the transferabilty of such perturbations in Sec. 6 and their robustness against defenses in Supp. Mat.

## 3 Finding Targeted Adversarial Perturbations

Given a pretrained depth prediction network, $f_d : \mathbb{R}^{H \times W \times 3} \to \mathbb{R}_+^{H \times W}$, $f_d : x \mapsto d(x)$ our goal is to find a small additive perturbation $v(x) \in \mathbb{R}^{H \times W \times 3}$, as a function of the input image $x$, which can change its prediction to a target depth $f_d(x + v(x)) = d^t(x) \neq d(x)$ with some norm constraint $||v(x)||_\infty \leq \xi$ and high probability $\mathbb{P}(f_d(x + v(x)) = d^t(x)) \geq 1 - \delta$.

We begin by examining Dense Adversarial Generation (DAG) proposed by [42] for finding adversarial perturbations for the semantic segmentation task. The perturbations from DAG can be formulated as the sum of a gradient ascent term (that pushes the predictions away from those of the original image) and a gradient descent term (that pulls predictions towards the target predictions). In the case of semantic segmentation, this formulation works well because the gradient ascent term suppresses the probability for the original predictions, which naturally increases the probability of the target predictions (zero-sum) driven by the gradient descent term. However, such is not the case for regression tasks, which requires the network to predict a real-valued scalar (as opposed to probability mass) for a targeted scene. Hence, the gradient ascent term maximizes the difference between the original and predicted depth, which results in DAG "overshooting" the target depth.

Instead, we use a simple objective function, similar to [17], but we modify it for the regression task by minimizing the normalized difference between predicted and target depth,

$$\ell(x, v(x), d^t(x), f_d) := \frac{||f_d(x + v(x)) - d^t(x)||_1}{d^t(x)}. \tag{1}$$

We minimize this objective with respect to an image $x$ by following an iterative optimization procedure (Alg. 1). The $\text{CLIP}\big(v_n(x), -\xi, \xi\big)$ operation clamps any value of $v(x)$ larger than $\xi$ to $\xi$ and any value smaller than $-\xi$ to $-\xi$. For all the experiments, $\xi \in \{2 \times 10^{-3}, 5 \times 10^{-3}, 1 \times 10^{-2}, 2 \times 10^{-2}\}$.

### 3.1 Implementation Details

We evaluate adversarial targeted attacks on KITTI semantic split [1]. This is a dataset of 200 outdoor scenes, captured by car-mounted stereo cameras and a LIDAR sensor, with ground-truth semantic segmentation and instance labels. The semantic and instance labels in this split enables our experiments in Sec. 5 for targeting specific categories or instances in a scene.

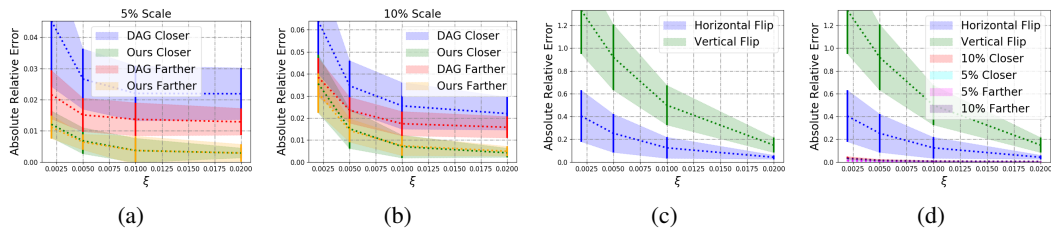

(a)            (b)            (c)            (d)

Figure 2: **ARE with various upper norm $\xi$ for scaling and flipping Monodepth2 predictions.** (a) and (b) Comparisons between DAG and the proposed method for scaling the scene by $\pm 5\%$ and $\pm 10\%$. (c) Results for horizontally and vertically flipping the predictions. (d) comparison between scaling and flipping tasks. Vertically flipping proves to be the most challenging.

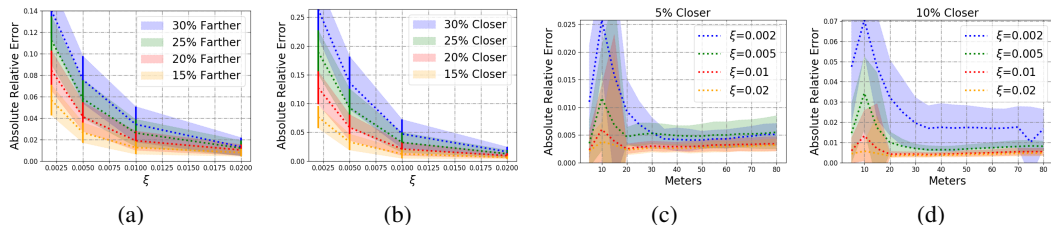

(a)            (b)            (c)            (d)

Figure 3: **Assessing larger scaling factors and error at various distances.** (a, b) ARE with various upper norm $\xi$ for scaling up to 30% closer and farther. With $\xi = 2 \times 10^{-2}$, an adversary to scale the scene up to 30% with little error. (c, d) ARE binned every 5m for scaling 5% and 10% closer. Most errors are concentrated around 10m. Note: scaling 5% and 10% farther also have similar plots.

The depth models (Monodepth, Monodepth2) are trained on the KITTI dataset [12] using Eigen split [6]. The Eigen split contains 32 out of the total 61 scenes, and is comprised of 23,488 stereo pairs with an average size of $1242 \times 375$. Images are resized to $640 \times 192$ as a preprocessing step and perturbations are computed with 500 steps of SGD. Entire optimization for each frame takes $\approx 12s$ (Monodepth2 takes 22ms for each forward pass and $11s \approx 500 \times 22ms$ in total) using a GeForce GTX 1080. Details on hyper-parameters are provided in the Supp. Mat.

For all the experiments, we use absolute relative error (ARE), computed with respect to the target depth $d^t(x)$, as our evaluation metric:

$$\text{ARE} = \frac{||f_d(x + v(x)) - d^t(x)||_1}{d^t(x)}. \tag{2}$$

## 3.2   Summary of Findings

By studying targeted adversarial perturbations and their effect on monocular depth prediction networks, we found (i) monocular depth prediction networks heavily rely on edges to determine scale of the scene (Sec. 4.1); (ii) with large perturbations (but still imperceptible), an adversary can fool the network to predict scenes completely different from the ones observed (Sec. 4.2, 4.3); (iii) depth prediction networks exhibit heavy biases present in the dataset (Sec. 4.2, 4.3); (iv) regions belonging to different semantic categories demonstrate different levels of robustness against targeted attacks (Sec. 5.1); (v) depth prediction networks leverage global context for local predictions, making them susceptible to non-local attacks (Sec. 5.2, 5.3); (vi) targeted perturbations do not transfer between networks, but, when crafted for multiple networks, can fool all of them equally well (Sec. 6).

## 4   Attacking the Entire Scene

Given a depth network $f_d$, our goal is to find adversarial perturbations to alter the predictions to a target scene $d^t(x)$ for an image $x$. For this, we examine three settings (i) scaling the entire scene by a factor, (ii) symmetrically flipping the scene, and (iii) altering the scene to a preset scene.

### 4.1   Scaling the Scene

For autonomous navigation, misjudging an obstacle to be farther away than it is could prove disastrous. Hence, to examine the possibility of altering the distances in the predicted scene without changing the scene topology or structure, we study perturbations that will scale the scene (bringing the scene closer to or farther away from the camera) by a factor of $1 + \alpha$. The target scene is defined as:

$$d^t(x) = \texttt{scale}(f_d(x), \alpha) = (1 + \alpha) f_d(x) \tag{3}$$

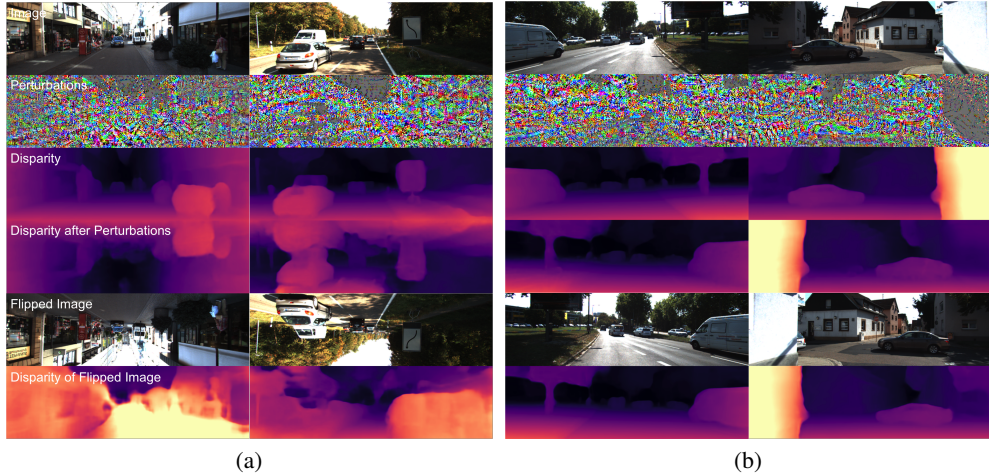

(a)                                         (b)

Figure 4: (a) **Examples of success (left) and failure (right) cases for vertical flip**. For the failure case, the car and road still remain on the bottom of the predictions. This is likely because the network is biased to predict closer structures on the bottom half of the image and farther ones on the top half (last two rows). (b) **Examples of horizontal flip.** Here, we observe the noise required to create and remove surfaces. Surprisingly, removing the white wall (right) requires very little perturbations.

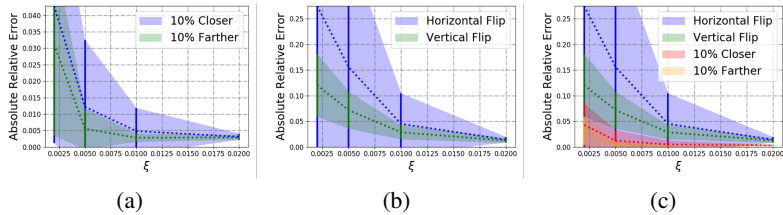

(a)                          (b)                          (c)

Figure 5: **ARE with various upper norm $\xi$ for scaling and flipping VNL predictions on NYUv2**. (a) Results for horizontally and vertically flipping the predictions, (b) Results for scaling the scene by $\pm 10\%$, and (c) comparison between scaling and flipping tasks. Unlike Monodepth2 trained on KITTI, VNL trained on NYUv2 is less biased by the orientation of the scene.

for $\alpha \in \{-0.10, -0.05, +0.05, +0.10\}$ or $-10\%$, $-5\%$ (closer), $+5\%$, $+10\%$ (farther), respectively. Column two of Fig. 1 shows the scene scaled 10% closer to the camera by applying visually imperceptible perturbations with $\xi = 2 \times 10^{-2}$. On average, scaling the scene by $-10\%$, $-5\%$, $+5\%$, $+10\%$ with $\xi = 2 \times 10^{-2}$ require an $\|v(x)\|_1$ of 0.0160, 0.0124, 0.0126, and 0.0161, respectively. We note that scaling the scene by $\pm 5\%$ requires less perturbations than $\pm 10\%$ and the magnitude required for both directions is approximately symmetric. Also, perturbations are typically located along the object boundaries with concentrations on the road. For a side by side visualization of comparisons between different scaling factors, please see the Supp. Mat.

In Fig. 2-(a, b), we compare our approach with DAG (Sec. 3), re-purposed for depth prediction task. While both are bounded by the same upper norm, DAG consistently produces results with higher error and generally with a higher standard deviation. As seen in Fig. 2-(a, b), even with $\xi = 2 \times 10^{-3}$, we are able to find perturbations that can scale the scene to be $\approx 1\%$ from $\pm 5\%$ and $\approx 3\%$ from $\pm 10\%$. With $\xi = 2 \times 10^{-2}$, we are able to fully reach all four targets with less than $\approx 0.5\%$ error.

We also consider larger scaling factors (up to 30% closer and farther) in Fig. 3-(a, b). While an adversary using the smallest upper norm, $\xi = 2 \times 10^{-3}$, is unable to alter the scene to the desired scale, one using the largest, $\xi = 2 \times 10^{-2}$, can still achieve target with little error. For a more detailed discussion on larger scaling factors and limitations, please refer to the Supp. Mat. To examine where the errors generally occur, we binned ARE every 5 meters (m) in Fig. 3-(c,d) and found that the error is generally concentrated around 10 meters.

## 4.2 Symmetrically Flipping the Scene

We now examine the problem setting where the target scene still retains the same structures given by the image, however, they are mirrored across the y-axis (horizontal flip) or x-axis (vertical flip).

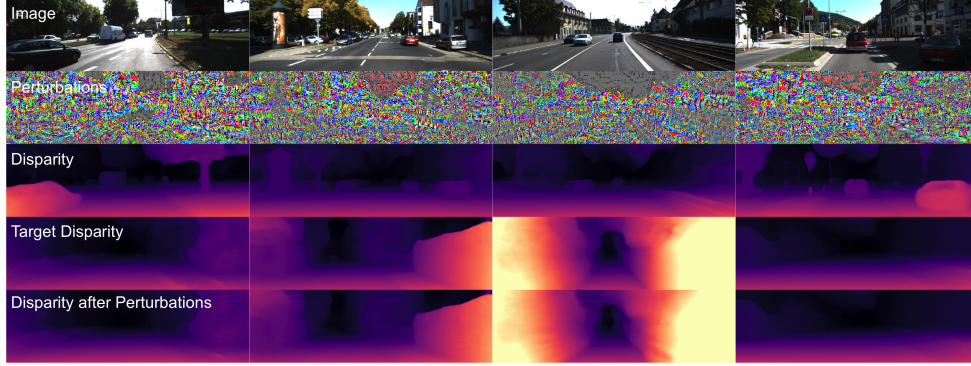

Figure 6: **Altering the predicted scene to a preset scene.** Adversarial perturbations can remove a car and replace a road sign with trees (leftmost), add walls (column two), and vegetation (column three) to open streets, and transform an urban environment with vehicles to an open road (rightmost).

For the *horizontal flip* scenario, we denote the target depth as $d^t(x) = \mathtt{fliph}(f_d(x))$ where the `fliph` operator horizontally flips the predicted depth map $f_d(x)$ across the y-axis.

Fig. 4-(b) shows that the perturbations can fool the network into predicting horizontal flipped scenes. For scenes with different structures on either side, $v(x)$ fools the network into *creating and removing* surfaces. Note that the amount of noise required to horizontally flip the scene is much more than that to scale the scene (i.e. for $\xi = 2 \times 10^{-2}$, $\|v(x)\|_1 = 0.0161$ for scaling $+10\%$, and $\|v(x)\|_1 = 0.0326$ for horizontal flip), which illustrates the difficulty in altering the scene structures. Interestingly, the amount of noise required to remove the white wall (Fig. 4-(b)) is significantly less than the rest.

For the *vertical flip* scenario, we denote the target depth as $d^t(x) = \mathtt{flipv}(f_d(x))$ where `flipv` operator vertically flips the predicted depth map $f_d(x)$ across the x-axis.

As seen in Fig. 4-(a), perturbations cannot fully flip the predictions vertically. Even on successful attempts (left), there are still artifacts in the output. For failure cases (right), portions of the cars still remain on the bottom half of the predictions. This experiment reveals the potential *biases* learned by the network. To verify this, we feed vertically flipped images to the network. As seen in the last two rows of Fig. 4-(a), the network still assigns closer depth values to the bottom half of the image (now sky) and farther depth values to the top half (now road and cars). Unlike outdoor datasets, indoor ones do not exhibit this orientation bias. Fig. 5 shows that it is in fact easier to fool VNL [44], trained on NYUv2 [36], into vertically flipping the scene than it is to achieve a horizontal flip.

In Fig. 2-(d) and Fig. 5-(c), we plot the ARE achieved by the proposed method for different methods and target depth maps: horizontal flip, vertical flip and different scales. Both flipping tasks are much harder than the scaling tasks. Especially for outdoor settings, fooling the network to produce vertical flipped predictions is the most challenging task as the error is $\approx 15\%$, even with $\xi = 2 \times 10^{-2}$.

### 4.3 Altering Predictions to Fit a Preset Scene

We now examine perturbations for altering the predicted scene $d(x_1) = f_d(x_1)$ to an entirely different pre-selected one $d^t(x_1) = f_d(x_2)$ obtained from images sampled from the same training distribution $x_1, x_2 \sim \mathbb{P}(x)$: $d(x_1) = f_d(x_1) \neq d^t(x_1) = f_d(x_2)$.

Fig. 6 shows that cars can be removed and road signs can be replaced with trees (leftmost), walls (column two) and vegetation (column three) can be added to the scene, and an urban street with vehicles can be transformed to an open road (rightmost). While perturbations are visually imperceptible, we note that $\|v(x)\|_1 = 0.0348$ (similar to that required for flipping). Even with this amount of noise, an adversary can fool the network to return scenes that are different from the one observed.

Additionally, this experiment also confirms the biases learned by network discussed in Sec. 4.2. While perturbations can alter the scene to a preset one with *structures not present* in the image, we have difficulties finding perturbations that can vertically flip the predicted scene.

## 5 Localized Attacks on the Scene

Given semantic and instance segmentation [1], we now examine adversarial perturbations to target localized regions in the scene. Our goal is to fool the network into (i) predicting depths that are closer or farther by a factor of $1 + \alpha$ for all objects belonging to a semantic category, (ii) removing specific

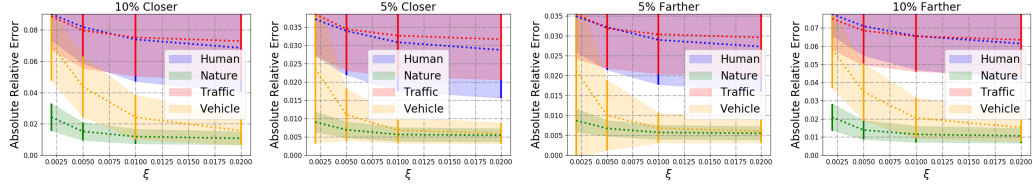

Figure 7: **ARE for scaling different categories closer and farther.** It is easier to fool the network to predict vehicle and nature categories closer and farther than is to fool human and traffic categories.

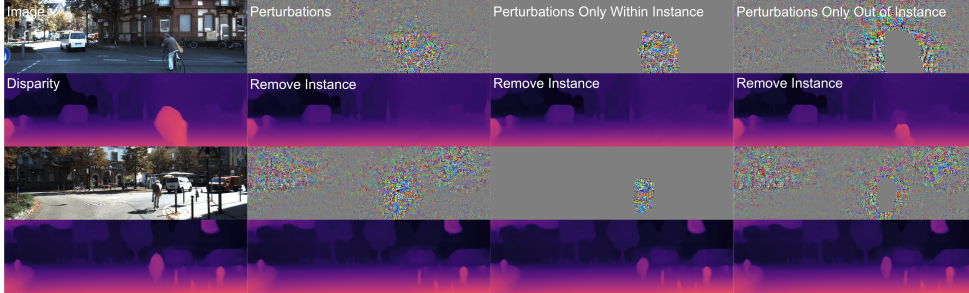

Figure 8: **Selectively removing instances of human (bikers and pedestrians).** Removing a localized target requires attacking non-local contextual information. Moreover, one can attack an instance without perturbing it at all. We demonstrate this by constraining the perturbation to be either completely within the instance mask or completely out of the mask.

instances from the scene, and (iii) moving specific instances to different regions of the scene, all the while keeping the rest of the scene unchanged.

## 5.1 Category Conditioned Scaling

Unlike Sec. 4.1, we want to alter a subset of the scene, partitioned by semantic segmentation, such that predictions belonging to an object category (e.g. vehicle, nature, human) are brought closer to or farther from the camera by a factor of $1 + \alpha$ for $\alpha \in \{-0.10, -0.05, +0.05, +0.10\}$.

We assume a binary category mask $M \in \{0, 1\}^{H \times W}$ derived from a semantic segmentation where all pixels belonging to a category are marked with $1$ and $0$ otherwise. We denote the target depth as

$$d^t(x) = (\mathbf{1} - M) \circ f_d(x) + (1 + \alpha)M \circ f_d(x) \tag{4}$$

where $\mathbf{1}$ is an $H \times W$ matrix of 1s. Column three of Fig. 1 illustrates this problem setting where the perturbations fool Monodepth2 into predicting all vehicles to be 10% closer to the camera. Fig. 7 shows a comparative study between different categories. Unlike Sec. 4.1, it is more difficult to alter a specific portion of the scene without affecting the rest. We surmise that this is due to the network's dependency on global context and hence altering one portion also affects other regions.

Moreover, each category exhibits a different level of robustness to adversarial noise. *Some categories are harder to attack than others*, e.g. traffic signs and human categories ($\approx 3\%$ error for $\alpha = \pm 5\%$ and $\approx 6\%$ for $\alpha = \pm 10\%$) are harder to alter than vehicle and nature ($\approx 0.5\%$ error for $\alpha = \pm 5\%$ and $\approx 1\%$ for $\alpha = \pm 10\%$). For interested readers, please see Supp. Mat. for visualizations, additional experiments, and performance comparisons amongst all categories.

## 5.2 Instance Conditioned Removing

We now consider the case where instance labels are available for removing a specific instance (e.g. car, pedestrian) from the scene. By examining this scenario, we hope to shed light on the possibility that a depth prediction network can "miss" a human or car, which may cause incorrect rendering in augmented reality or an accident in the autonomous navigation scenario.

Similar to Sec. 5.1, we assume a binary mask $M$, but in this case, of specific instance(s) in the scene, e.g. all pixels belonging to a specific pedestrian are marked with $1$ and $0$ otherwise. To obtain $d^t(x)$, we first remove the depth values in $f_d(x)$ belonging to $M$ by multiplying $f_d(x)$ by $\mathbf{1} - M$. Then, we use the depth values $f_d(x)$ on the contour of $M$ to linearly interpolate the depth in the missing region:

$$d^t(x) = (\mathbf{1} - M) \circ f_d(x) + M \circ d_M^t(x) \tag{5}$$

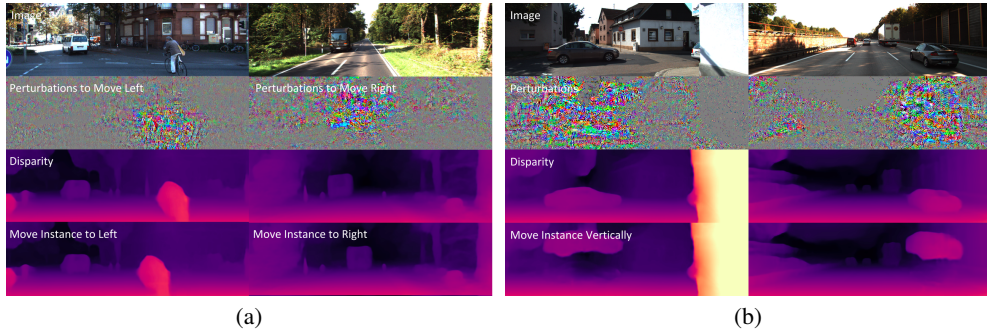

(a)                                         (b)

Figure 9: (a) **Moving horizontally.** Selected instances is moved by $\approx 8\%$ in the left and right directions while rest of the scene is preserved. (b) **Flying cars.** A vehicle instance is moved $\approx 42\%$ upward while rest of the scene is preserved. Noise is concentrated around the targeted instance.

where $d_M^t(x) := \mathtt{interp}(\mathtt{contour}(f_d(x), M))$. Indeed, this scenario is possible. Fig. 8 shows examples of pedestrian and biker removal in the driving scenario where perturbations completely remove the targeted instance. With this attack, the road ahead becomes clear, which makes the agent susceptible to causing an accident.

Even though perturbations are concentrated on the targeted instance region, non-zero perturbations can be observed in the surrounding regions. While the target effect is localized (e.g., make a pedestrian disappear), the perturbation is non-local, implying that the network exploits non-local context, which presents a vulnerability to attacks against a target instance by perturbing other parts of the image. To validate this claim, in the next section, we study perturbations with spatial constraints (either within or outside of the mask) to restrict the information that the adversary can attack.

### 5.3 Instance Conditioned Removing with Spatial Constraints on the Perturbations

Motivated by our results in Sec. 5.2, we extend the instance conditioned removal task to more constrained scenarios where the perturbations have to either exist (i) completely within the target instance mask $M$ or (ii) completely outside of it.

For perturbations within the targeted instance mask $M$, we constrain $v(x)$ to satisfy $||M \circ v(x)||_\infty \leq \xi$ and $||(\mathbf{1} - M) \circ v(x)||_\infty = 0$. When constrained within $M$, perturbations can only remove *some instances* successfully (e.g. biker is completely removed in row two, column three of Fig. 8). In other cases, the perturbations can only remove the outer part of the instance, leaving parts of the instance in the scene (row four). This shows that depth prediction networks leverage global context; without attacking the contextual information located outside of $M$ (e.g. without perturbing the entire image as in column two of Fig. 8), it is not always possible to completely remove the target instance.

Second, we want to answer the question posed in Sec. 1. Can perturbations remove an object by attacking anywhere (e.g. a billboard), *but the object* (e.g. a car)? In this more challenging case, the perturbations are constrained to be outside of the instance mask: $||(\mathbf{1} - M) \circ v(x)||_\infty \leq \xi$ and $||M \circ v(x)||_\infty = 0$. Column four of Fig. 8 shows that even though there are no "direct attacks on the object" (perturbations in the masked region), the perturbations can *still remove parts of the target instance*. While some of the target instance still remains, our experiment demonstrates that depth prediction networks are indeed susceptible to *attacks against a target instance that does not require perturbing the instance at all*.

### 5.4 Instance Conditioned Translation

In this case study, we examine perturbations for moving an instance (e.g. vehicle, pedestrian) horizontally or vertically in the image space. As Sec. 5.2 and 5.3 have demonstrated the ability to remove localized objects from the scene, we now show that it is possible for perturbations to move such objects to different locations in the scene (removing the instance and creating it elsewhere) while keeping the rest of the scene unchanged.

Fig. 9-(a) shows that perturbations can fool a network to move the target instance by $\approx 8\%$ across the image in the left and right directions. When moved left, the biker (left column) is now in front our vehicle. When moved right, the truck (right column) is in the wrong lane and looks to be on-coming traffic. Moreover, Fig. 9-(b) shows that perturbations can move select instances by $\approx 42\%$ in the upward direction, creating the illusion that there are "flying cars" in the scene.

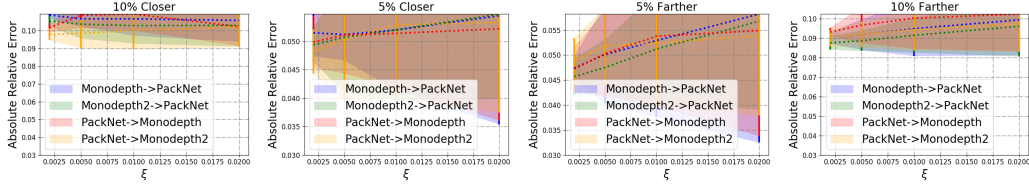

Figure 10: **Transferability between networks with different architectures and loss functions.** Perturbations are optimized for Monodepth, Monodepth2 (both with 2D convolutions) and PackNet (3D convolutions) separately and are transferred from Monodepth models to PackNet and vice versa.

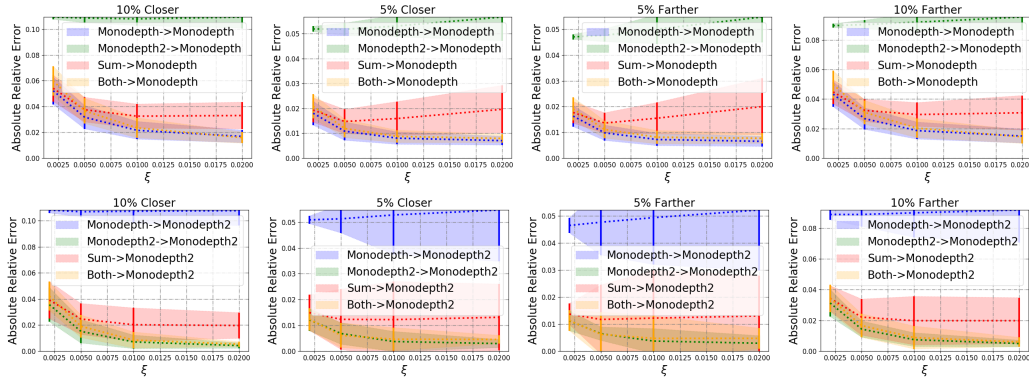

Figure 11: **Transferability between networks with similar architectures and loss functions.** Perturbations are (i) optimized for Monodepth and Monodepth2 separately, (ii) optimized for both together and (iii) summed over perturbations calculated for Monodepth and Monodepth2 separately.

## 6 Transferability Across Different Models

Transferability is important for black-box scenarios, a practical setting where the attacker does not have access to the target model or its training data. To examine transferability, we test our perturbations crafted for Monodepth2 [14] to fool its predecessor Monodepth [13] (both use 2D convolutions and similar loss functions) in Monodepth2→Monodepth, and vice versa in Monodepth→Monodepth2, for the scene scaling task. (Sec. 4.1). To consider networks with different architectures and loss functions, we also test perturbations crafted for Monodepth and Monodepth2 on PackNet [16] (3D convolutions) in Fig. 10.

To this end, we also optimized perturbations for Monodepth and PackNet to scale the entire scene. Overall, the perturbations optimized for one model does not transfer to another. Fig. 10 shows that perturbations crafted for networks with different architectures do not transfer. When comparing models with similar architectures and loss, interestingly, transferability decays with increasing norm (Fig. 11), which may be due to perturbations overfitting to the model. We summed the perturbations for Monodepth and Monodepth2 ("Sum" in Fig. 11) and found that their summation can *affect both models* with reduced effects as the upper norm increases. For $\xi = 2 \times 10^{-3}$, the potency is nearly unaffected, meaning, for small norms, their summation can attack both models equally well. Lastly, by optimizing for both models ("Both" in Fig. 11), the *same perturbation* can fool both as if it was optimized for the models individually, with performance indistinguishable from Monodepth→Monodepth and Monodepth2→Monodepth2 *across all norms*. This shows that both models share a space that is vulnerable to adversarial attacks. Hence, crafting perturbations for an array of potential models may be an avenue towards achieving absolute transferability across models.

## 7 Conclusion

Depth prediction networks are indeed vulnerable to adversarial perturbations. Not only can such perturbations alter the perception of the scene, but can also affect specific instances, making the network behave unpredictably, which can be catastrophic in applications that involve interaction with physical space. These perturbations also shed light on the network's dependency on non-local context for local predictions, making non-local targeted attacks possible. Despite these vulnerabilities, targeted attacks do not transfer and cannot be crafted in real time, so we are safe for now. Yet, the fact that they exist suggests that there is room to improve the representation learned. We hope that our findings on the network's biases, the effect of context, and robustness of the perturbations can help design more secure and interpretable models that are not susceptible to such attacks.

## Broader Impact

Adversarial perturbations highlight limitations and failure modes of deep networks. They have captured the collective imagination by conjuring scenarios where AI goes awry at the tune of imperceptible changes. Some popular media and press has gone insofar as suggesting them as proof that AI cannot be trusted.

While monocular depth prediction networks are indeed vulnerable to these attacks, we want to assure the reader that these perturbations cannot cause harm outside of the academic setting. As mentioned in Sec. 3.1, optimizing for these perturbations is computationally expensive and hence it is infeasible to craft these perturbations in real time. Additionally, they also do not transfer; so, we see little negative implications for real-world applications. However, the fact that they exist implies that there is room for improvement in the way that we learn representations for depth prediction.

Hence, we see the existence of adversaries as an opportunity. Studying their effects on deep networks is also the first step to render a system robust to such a vulnerability. The broader impact of our work is to understand the corner cases and failure modes in order to develop more robust representations. This, in turn, will improve interpretability (or, rather, reduce nonsensical behavior). In the fullness of time, we expect this research to pay a small contribution to benefit transportation safety.

## Acknowledgements

This work was supported by ONR N00014-19-1-2229 and ARO W911NF-17-1-0304.

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
