[Supplementary Material]

# Targeted Adversarial Perturbations
# for Monocular Depth Prediction
# SUPPLEMENTARY MATERIALS

**Alex Wong**
UCLA Vision Lab
alexw@cs.ucla.edu

**Safa Cicek**
UCLA Vision Lab
safacicek@ucla.edu

**Stefano Soatto**
UCLA Vision Lab
soatto@cs.ucla.edu

## 1 Summary of Contents

In Sec. 2, the robustness of perturbations against defenses is discussed. Additional implementation details that we could not fit into main text due to space constraints are given in Sec. 3. We verify our claim that targeted adversarial perturbations are visually imperceptible in Sec. 4. More experimental results on changing the scale of the scene are provided in Sec. 5. In Sec. 6, existence of the successful adversarial attacks for indoor scenes (NYU-V2) is shown for state-of-the-art indoor monocular depth prediction model. In Sec. 7, we examine how predictions behave when linear operations are applied to perturbations (sum of two perturbations and linear scaling of a perturbation). Failure cases for the perturbations are analyzed in Sec. 8. Finally, in Sec. 9, more qualitative and quantitative results are provided for the experiments whose compressed versions are presented in the main text.

## 2 Robustness of the Targeted Attacks Against Defense Mechanisms

In the main text, we have shown that depth prediction networks are prone to adversarial attacks. In this section, we will examine the robustness of the perturbations against common defense mechanisms: (i) Gaussian blurring and (ii) adversarial training.

### 2.1 Defense through Gaussian Blurring

In Fig. 1, we show the effect of Gaussian blurring as a simple defense mechanism on our targeted attacks by blurring the image with additive perturbations. Even though Gaussian blur does reduce the effectiveness of the perturbations (increased ARE over all scales), the resulting scene is still only $\approx 3\%$ away from a target depth that is 10% closer or farther than the original predictions for $\xi = 2 \times 10^{-2}$. This is the performance that the method achieves for the case of $\xi = 2 \times 10^{-3}$ without blurring. In other words, the effect of the blurring can be suppressed simply by increasing the upper norm of the noise by $10\times$ for scaling the scene by $\pm 10\%$.

Figure 1: **Gaussian blur.** Absolute relative error (ARE) achieved by adversarial perturbations of different norms ($\xi$) for different scales. For each scale, we plot the ARE with and without Gaussian blur. Even though absolute relative error increases with the Gaussian blur, the proposed method can still find a small norm noise to alter the scene.

Figure 2: **Adversarial training.** Absolute relative error (ARE) achieved by adversarial perturbations of different norms ($\xi$) for different scales. For each scale, we plot the ARE with and without adversarial training. Even though absolute relative error increases with the adversarial training, the perturbations can still affect the predicted scene with small norm noise.

Figure 3: **Adversarial training vs Gaussian blur.** Absolute relative error (ARE) achieved by adversarial perturbations of different norms ($\xi$) for different scales. For each scale, we plot the ARE without any defense, with Gaussian blur and with adversarial training. Both Gaussian blur and adversarial training makes the depth prediction network more robust to perturbations. Performances of the two defense mechanisms are comparable for small norms ($\xi$), but adversarial training is more effective on larger norms.

## 2.2 Defense through Adversarial Training

To examine the robustness of adversarial perturbations to adversarial training, we crafted adversarial perturbations for scaling the scene by a factor of $1 + \alpha$ where $\alpha \in \{-0.10, -0.05, +0.05, +0.10\}$ for the KITTI Eigen split [3] (consisting of 22600 stereo pairs). We trained Monodepth2 [5] by minimizing the normalized discrepancy between the predicted depth of a perturbed image ($f_d(x + v(x))$) and its prediction for the original image ($f_d(x)$).

$$\ell(x, v(x), f_d) = \frac{\|f_d(x) - f_d(x + v(x))\|_1}{f_d(x)} \tag{1}$$

Fig. 2 shows the effect of the perturbations on Monodepth2 after adversarial training. While training does reduce the influence of adversarial perturbations on the scene scaling task, it does not make the network invariant to the adversarial perturbations. With perturbations of $\xi = 2 \times 10^{-3}$, the predicted scene is $\approx 7\%$ from the target scene that is $10\%$ closer than or farther from the original and $\approx 3\%$ from the target scene that is $5\%$ closer or farther. For $\xi = 2 \times 10^{-2}$, perturbations can still fool the network to be predict a target scene scaled by $\pm 10\%$ with $\approx 5\%$ absolute relative error and $\approx 2\%$ error for fooling the network to predict a target scene that is scaled by $\pm 10\%$.

To compare the two defense mechanisms, we refer to Fig. 3. For smaller norms, e.g. $\xi = 2 \times 10^{-3}$, we observe a similar performance in using Gaussian blur (Sec. 2.1) and adversarial training as defenses against adversarial perturbations; whereas, adversarial training is clearly better for larger $\xi$. This may be due to Gaussian blur's ability to destroy the perturbation for small norms and, hence, able to mitigate the effect of the perturbations. However, for larger norms, the blurring does not corrupt the perturbations enough and therefore does not reduce the effect of perturbations by as much.

## 3 Additional Implementation Details for Outdoor Scenario

In this section, we provide the additional implementation details for crafting adversarial perturbations for Monodepth [4] and Monodepth2 [5] on the KITTI dataset (outdoor driving scenario) as discussed in the main text.

| Upper Norm | $\xi = 2 \times 10^{-3}$ | $\xi = 5 \times 10^{-3}$ | $\xi = 1 \times 10^{-2}$ | $\xi = 2 \times 10^{-2}$ |
|---|---|---|---|---|
| Monodepth | $\eta = 1.0$ | $\eta = 2.0$ | $\eta = 3.0$ | $\eta = 4.0$ |
| Monodepth2 | $\eta = 0.1$ | $\eta = 1.0$ | $\eta = 3.0$ | $\eta = 5.0$ |

Table 1: **Learning rates.** We achieve the best performances with the given learning rates.

### 3.1 Hyper-parameters

Regarding hyper-parameters for crafting *adversarial perturbations*: We search the learning rate for each noise norm from the set $\{0.1, 1.0, 2.0, 3.0, 4.0, 5.0, 10.0\}$. We report the best performing ones in Table 1. Regarding our choice for the number of steps to run, we experimented with 200, 400, 500, 800, and 1000 steps and found little difference in performance measured by ARE between 500 steps, and 800 and 1000 steps. While an increased number of steps will obtain slight performance improvements, conscious of the time complexity, we chose 500 for our experiments.

Regarding hyper-parameters for *adversarial training*: As a defense against adversarial perturbations, we optimized Eqn. 1 for Monodepth2 using Adam [7] with $\beta_1 = 0.9$ and $\beta_2 = 0.999$. We used a batch size of 4 and starting learning rate of $1 \times 10^{-5}$. We decreased the learning rate to $5 \times 10^{-6}$ after 10 epochs and to $2.5 \times 10^{-6}$ after 20 epochs and $1 \times 10^{-6}$ after 30 epochs for a total of 40 epochs. Training takes approximately 4 hours using an Nvidia GeForce GTX 1080 GPU.

### 3.2 Monodepth, Monodepth2, and PackNet

We study the effects of adversarial perturbations on the state-of-the-art monocular depth prediction method, PackNet [6], Monodepth2 [5] and its predecessor Monodepth [4]. Monodepth2 and Monodepth models utilize similar 2D convolutional network architectures and are also trained with similar loss functions. PackNet is built on 3D convolutions and uses a different loss function than Monodepth models. In this section, we provide details on the three methods.

Regarding *Monodepth*: Monodepth uses a ResNet50 encoder architecture as its backbone and a standard decoder with skip connections. Monodepth predicts both left and right disparities from a single image (assuming it is the left image of a stereo-pair) and uses image reconstruction as supervision. Additionally, it is trained with a standard local smoothness term weighted by image gradients and a left-right disparity consistency term as its regularizers.

Regarding *Monodepth2*: Monodepth2, unlike Monodepth, uses ResNet18 encoder (pretrained on ImageNet) as its backbone network architecture. Rather than simply minimizing an image reconstruction loss, Monodepth2 leverages a heuristic to discount occluded pixels and also uses a criterion to discount static frames. Similar to Monodepth, Monodepth2 also minimizes a local smoothness regularizer weighted by image gradients.

Regarding *PackNet*: PackNet uses 3D convolutions. While the general architecture is still of the encoder-decoder form, they use "packing" and "unpacking" to convert 2D features into 3D features. PackNet and the Monodepth methods all leverage structure from motion, but PackNet also uses velocity at every time step as a prior to obtain metric depth.

## 4 Imperceptibility

While we have provided the $L_\infty$ norm (from $2 \times 10^{-3}$ to $2 \times 10^{-2}$) and $L1$ norm (ranging from $\approx 0.0124$ to $\approx 0.0348$, depending on the target scene) of the perturbations in the main text, it is difficult to quantify how "visually imperceptible" these perturbations really are. The real test is whether a human can spot the difference between the original and perturbed image. Hence, we conducted a study with 28 people where we ask if two images (the original image paired with itself or its associated perturbed image with the highest norm, $\xi = 0.02$) are the same. The perturbation types were randomly chosen from scaling $\pm 10\%$, horizontal, or vertical flipping. Of the 40 random pairs (20 perturbed), users identified only 1 perturbed image on average, which verifies our claim on the imperceptibility of the perturbations.

Figure 4: **ARE with various upper norm $\xi$ for scaling Monodepth2 predictions by larger factors.** We increased the scaling to $15\%$, $20\%$, $25\%$ and $30\%$ closer and farther. We can see that for $\xi = 2 \times 10^{-2}$, perturbations are still able to scale the entire scene by $\approx 1\%$ error.

Figure 5: **Indoor quantitative results.** ARE with various upper norm $\xi$ for scaling and flipping VNL predictions. (a) Results for horizontally and vertically flipping the predictions. (b) Results for scaling the scene by $\pm 10\%$. (c) comparison between scaling and flipping tasks.

## 5   Scaling with Larger Factors

In Sec. 4.1 and Fig. 2-(a, b) of the main text, we showed that it is possible, even for small norms such as $\xi = 2 \times 10^{-3}$ to scale the scene to be $5\%$ or $10\%$ closer or farther with small error. In this section, we demonstrate that it is possible to scale the scene by larger amounts (up to $30\%$ closer or farther). Fig. 4 shows that with $\xi = 2 \times 10^{-3}$, it is only able to scale the up to $15\%$ with reasonable error; whereas perturbations with $\xi = 5 \times 10^{-3}$ can achieve this up to $20\%$ closer or farther. However, using larger norms ($1 \times 10^{-2}$ and $\xi = 2 \times 10^{-2}$), one can scale the scene up to $30\%$ with small errors (less than $5\%$ ARE for $\xi = 1 \times 10^{-2}$ and $\approx 1\%$ ARE for $\xi = 2 \times 10^{-2}$).

To see how far we can push for each upper norm, Fig. 9 shows various scales that each upper norm is capable of achieving. We note that $\xi = 2 \times 10^{-2}$, is still able to obtain less than $2\%$ ARE when scaling the scene by $45\%$; however, standard deviation starts to grow larger as the scaling increases.

## 6   Adversarial Attacks for Indoor Scenes

To show the applicability of the adversarial method on indoor scenes, we examine the adversarial perturbations for Kinect Dataset NYU Depth V2 (NYU-V2) [8]. We tested the effectiveness of adversarial perturbations on Virtual Normal Loss (VNL) [10] which is the state-of-the-art monocular depth prediction method for NYU-V2, trained in the supervised setting.

### 6.1   Implementation Details

NYU-V2 consists of 1449 RGBD images gathered from a wide range of buildings, comprising 464 different indoor scenes across 26 scene classes. The images were hand-selected from 435,103 video frames, to ensure diversity. 1449 labeled samples are split into 795 training and 654 test images.

In the method proposed by [10], a 3D point cloud is reconstructed from the estimated depth map. Then, three non-colinear points are randomly sampled with large distances to form a virtual plane. The

Figure 6: **Indoor qualitative results.** From left to right: (a) horizontal flip, (b) vertical flip, (c) scale 10% closer and (d) scale 10% farther. From top to bottom: RGB, noise, original disparity, disparity prediction for the perturbed image.

deviation between ground truth and prediction for the direction of the normal vector corresponding to the plane is penalized. The pre-trained ResNeXt-101 [9] model on ImageNet [2] is used as the backbone architecture. During training, images are cropped to the size $384 \times 384$ for NYU-V2. We use the same image resolution for our experiments. The training set is randomly sampled from 29K images of the raw unlabeled training set.

The time it takes to forward an image with this method is $\approx 0.15$ seconds ($\approx 7$ times more than Monodepth2). Due to computational limitations, we choose the first 20 images out of 654 images of the test split for our experiments. We run SGD for 1000 steps. The learning rate is kept at 10.0 for the entire optimization.

## 6.2 Scaling and Symmetrically Flipping the Scene

In Fig. 5, we compare the performance for different target depth maps: scaling the scene by $\pm 10\%$, horizontal and vertical flipping. Unlike outdoor case (KITTI), in the indoor (NYU-V2) horizontal flipping is a harder task than vertical flipping. Achieving a vertically flipped scene was expected to be simpler as layouts of indoor scenes are more diverse. Hence, the depth network does not overfit to

a particular layout type e.g. the one in which there are large depth values only at the top of the image. Horizontal flipping being relatively harder for indoor scenes can be explained by the large divergence between the depth distributions of the original predictions and the target depths. The reason is that for the indoor scenes, most scenes are not symmetric in the horizontal direction, unlike the outdoor driving scenario where the left and right parts of the scene from the ego-view are usually symmetric.

Since [10] normalizes images with the deviation of the dataset which approximately scales the image by 5, it also effectively scales the noise with the same deviation. But, since the relative norm is still the same, we use the same norm values $\xi \in \{2 \times 10^{-3}, 5 \times 10^{-3}, 1 \times 10^{-2}, 2 \times 10^{-2}\}$ when plotting the ARE in Fig. 5.

In Fig. 6, we present qualitative results for NYU-V2 for $\xi = 2 \times 10^{-2}$. Small, white borders around RGB images exist in the raw dataset. For all the tasks, including vertical flip, adversarial perturbations manage to fool the model to predict the target depth with small errors. For horizontally flipped target depth (a), predictions have more artifacts than vertical flipped depth (b) and scaled depths (c,d).

# 7 Linear Operations

Figure 7: Disparity for $x + \gamma v(x)$ where $\gamma$ is 0.0, 0.25, 0.5, 0.75, 1.0 from top to bottom. $v(x)$ is calculated for $d^t = \texttt{fliph}(f_d(x))$. So, the top is the original disparity map while bottom most is the flipped one. In between, portions of the scene are flipped smoothly.

Figure 8: (1st row) left to right: RGB, sum of noises. (2nd row) left to right: original disparity, disparity when two noises $v_1(x) + v_2(x)$ are added to the image. (3rd row) left to right: noise for 10% closer, noise for 10% farther. (4th row): Disparity predictions for the images perturbed with the noises in the 3rd row. When added, two noises cancel each other's effect: the scale for $f_d(x + v_1(x) + v_2(x))$ is close to the original one $f_d(x)$.

To better understand how predictions of the depth network changes within a ball of small radius, we examine the effect of linear operations on perturbations. Specifically, we visualize the predictions for the scaled perturbations and for the perturbations which we get after summing two perturbations calculated for two different target depth maps.

Figure 9: **ARE with various upper norm $\xi$ for scaling Monodepth2 predictions.** This time error is plotted for large scale ratios $15\%$, $20\%$, $25\%$ and $30\%$ (up to $45\%$ for larger $\xi$), for scaling both closer and farther, showing the limitations of each norm for the scaling task. While ARE is still relatively small for larger norms, standard deviation grows larger – meaning the perturbations can no longer scale the scene consistency with low error.

Figure 10: *Additional examples of failure cases for vertical flip.* While adversarial perturbations with $\xi = 2 \times 10^{-2}$, can fool Monodepth2 to predicted scenes that are scaled by large amounts, and horizontally flipped. They cannot cause the network to vertically flip the scene, leaving behind cars and roads as artifacts.

In Fig. 7, we take the perturbation $v(x)$, which we calculated to horizontally flip the prediction for the given image, and we visualize the prediction of the network for $x + \gamma v(x)$ where $\gamma \in \{0.0, 0.25, 0.5, 0.75, 1.0\}$. As can be seen, between $\gamma = 0$ and $\gamma = 1$, the scene is smoothly flipped. This implies that the adversarial perturbations can be used to control the depth prediction in a disentangled way. In other words, one causal factor (e.g. horizontal orientation) of the prediction can be independently controlled by tweaking $\gamma$ only, keeping everything else the same.

As observed before, noise is small for the white regions. See the third column, where there is a gray rectangle in the noise corresponding to the white region of the truck. We speculate the reason behind this phenomenon as the white color being on the border of the support of RGB images. But, the noise is still large for black regions which are at the other extreme of the support (see perturbations corresponding to black vehicles). So, we left further understanding of this phenomenon as future work.

In Fig. 8, we take $v_1(x)$ and $v_2(x)$ which are optimized to scale the scene to 10% closer and 10% farther. Then, visualize the summed perturbation, $v(x) = v_1(x) + v_2(x)$ and the prediction for $x + v(x)$. As can be seen, two noises cancel each other: $||v_1(x)|| \approx ||v_2(x)|| \gg ||v_1(x) + v_2(x)||$. Furthermore, the prediction for the image perturbed with the summed noise is close to the original prediction: $f_d(x) \approx f_d(x + v_1(x) + v_2(x))$. This shows that two perturbations with inverse functionalities can neutralize their effects when applied together.

## 8 Failure Cases

Fig. 9 shows the absolute relative error (ARE) with respect to the target scaling factor for each upper norm. As we can see, for smaller norms of $2 \times 10^{-3}$ and $5 \times 10^{-3}$, the perturbations are limited to scaling the scene by $\approx 15\%$ and $\approx 20\%$, respectively. Scaling factors higher than such increases the ARE by $\approx 4\%$ for every $5\%$ increase in scaling factor, signaling the limit for these norms. For larger norms of $1 \times 10^{-2}$ and $2 \times 10^{-2}$, the perturbations can afford to scale the scene by a much larger factor. For $\xi = 1 \times 10^{-2}$, perturbations can scale the scene by as much as $30\%$ closer and farther

less than $5\%$ error. Whereas, for $\xi = 1 \times 10^{-2}$, perturbations can scale the scene up to $\pm 45\%$ with less than $2\%$ ARE. However, while large scaling still as a low ARE, the standard deviation for larger norms increases drastically showing that it can no longer consistently scale the scene.

While for smaller scales (e.g. $\pm 5$, $\pm 10$) the ARE and amount of noise required is approximately the same (see Sec. 4.1, main text), suggesting similar difficulty levels. As we plot the errors for larger scales in Fig. 9, scaling the scene farther generally yields lower error than scaling the scene closer.

Fig. 10 shows additional examples of failure cases for vertical flip. While we have shown in the main paper as well as Sec. 5 and 9 that it is possible to manipulate the scene with small norm perturbations, we show here that perturbations cannot fool a network into vertically flipping the scene.

# 9 Additional Results on Outdoor Scenarios

In this section, we show (i) side by side visualization of the perturbations required to scale the scene, (ii) additional visualizations of perturbations to horizontally and vertically flip the scene, (iii) quantitative results on targeted attacks to semantic categories and (iv) qualitative results on targeted attacks to instances.

## 9.1 Scaling the Scene

Figure 11: Visually imperceptible perturbations $v(x)$, with $\xi = 2 \times 10^{-2}$, can fool Monodepth2 to predicted scenes that are 5% or 10% closer and also 5% or 10% farther.

Here, we show qualitative results for the task of scaling the scene (Sec. 4.1, main text) by a factor of $1 + \alpha$ where $\alpha \in \{-0.10, -0.05, +0.05, +0.10\}$. As seen in Fig. 11, the perturbations are successful in fooling state-of-the-art monocular depth prediction method, Monodepth2 [5], into predicting the scene 5% or 10% closer and also 5% or 10% farther. Additionally, the perturbations are concentrated in similar regions for scaling the scene 5% or 10% closer and for 5% or 10% farther as well. As noted in the main text, the amount of noise required for scaling the scene by $\pm 5\%$ are approximately the same, as is the amount for scaling the scene by $\pm 10\%$. This is visible in Fig. 11.

## 9.2 Symmetrically Flipping the Scene

In Sec. 4.2 and Fig. 3 in the main text, we demonstrated that adversarial perturbation can cause a monocular depth prediction network to predict a horizontally or vertically flipped scene. Here, we show additional qualitative results on the horizontal and vertical flipping tasks in Fig. 12 and Fig. 13.

Figure 12: *Additional examples of horizontal flip*. Adversarial perturbations with $\xi = 2 \times 10^{-2}$, can fool Monodepth2 to predicted scenes that horizontally flipped.

Figure 13: *Additional examples of vertical flip*. Adversarial perturbations with $\xi = 2 \times 10^{-2}$, can fool Monodepth2 to predicted scenes that vertically flipped. Even in these successful examples, there are still artifacts (ripples, wavy-ness) in the output.

We note that perturbations can cause the network to predict a horizontally flip scene, they have trouble fooling the network to predict a vertically flipped scene. This is unlike our findings in the indoor scenario (Sec. 6) as seen in Fig. 6 and 5. Fooling the network to vertically flip the scene is in fact *easier* than fooling it to horizontally flip the scene. This confirms the biases (roads on bottom, sky on top) that the network learned from the outdoor dataset that are not present in the indoor dataset.

## 9.3 Category Conditioned Scaling

In Sec. 5.1 and Fig. 5 of the main text, we showed category specific attacks to scale all objects belonging a given category to be a factor of $1+\alpha$ closer or farther where $\alpha \in \{-0.10, -0.05, +0.05, +0.10\}$. Here, we provide performance, measured in ARE, of adversarial perturbations crafted for each category. We use the same convention for grouping different classes into categories as the Cityscapes dataset [1], with the exception of the "Human" category, which includes the bicycles that the bikers are riding.

Fig. 14 shows a comparative study between different categories. Not all categories are equally easy to be fooled by the perturbations, some are more robust to adversarial attacks than others. As seen in Fig. 14, each category exhibits a different level of robustness to adversarial noise – "Human" and "Traffic" categories are the hardest to fool, "Construction", "Vehicle" and "Flat" are more susceptible, and "Sky" and "Nature" are the easiest to attack. Plots are cropped at the maximum error across different categories to enable comparison of difficulty in fooling different categories. We note that

attacking localized regions in the scene is considerably harder than attacking the entire scene. Fig. 4 shows that perturbations can attack the entire scene with small errors across various norms while Fig. 14 shows that, even with large norms, there are still errors ($\approx 2\%$ to $6\%$ ARE). We show visualizations for the "Construction", "Nature", and "Vehicle" categories in Fig. 15, 16, and 17 respectively.

Figure 14: **ARE for scaling different categories closer and farther.** From top to bottom: "Construction", "Flat", "Human", "Nature", "Sky", "Traffic", "Vehicle". From left to right: 10% closer, 5% closer, 5% farther, 10% farther. Y-axis is kept the same for the same scale, for making the comparison across categories possible. It is easier to fool the network to predict vehicle and nature categories closer and farther than is to fool human and traffic categories.

Figure 15: **Examples of targeted attacks on regions belonging to "Construction" category.**

Figure 16: **Examples of targeted attacks on regions belonging to "Nature" category.**

Figure 17: **Examples of targeted attacks on regions belonging to "Vehicle" category.**

Figure 18: **Additional examples of human removal.** Targeted adversarial perturbations can remove humans from the predicted scene. Rightmost panel shows that we can target multiple humans and remove them from the scene without affecting the remaining pedestrian on the right.

Figure 19: **Examples of vehicle removal.** Targeted adversarial perturbations can remove vehicles from the predicted scene. Rightmost panel shows that we can target a truck and a car on the right side and remove them. Note that the cars in the middle of the road still remain.

## 9.4 Instance Conditioned Targeted Attacks

In Sec. 5.2 and Fig. 6 in the main text, we show that, when given instance segmentation, adversarial perturbations can target specific instances and remove them from the scene and thus causing unforseen consequences. Fig. 18 shows additional examples of removing humans from the scene and Fig. 19 demonstrates that it is possible to remove vehicles from the scene as well. In the rightmost panel of Fig. 18, we show that it is possible to remove *some* pedestrians from the scene without affecting others. Similarly, in the rightmost panel of Fig. 19, we removed a truck and a car on the right side and left the cars in the center untouched – leaving this as *still* a plausible highway driving scenario.

In Sec. 5.4 and Fig. 7 in the main text, we show that perturbations can move an instance to another location in the scene (requires removing the instance from its original location and creating it in the new location). In this section, we give more visuals for the perturbations used for moving an instance (e.g. vehicle, pedestrian) horizontally or vertically in the image space while keeping the rest of the scene unchanged.

Fig. 20 shows that perturbations can fool a network to move the target instance by $\approx 8\%$ across the image in the left and right directions. Furthermore, Fig. 21 shows that perturbations can move select instances by $\approx 42\%$ in the upward direction, creating the illusion that there are "flying vehicles" in the scene. We note that in both cases, the perturbations are concentrated on the instance and the region

Figure 20: **Move instance horizontally.** Selected instance is moved by $\approx 8\%$ in left and right directions while rest of the scene is preserved. Noise and disparity for both directions are given. We note that the noise is concentrated around the targeted instance and the region to which the instance is moved.

Figure 21: **Flying vehicles.** Selected vehicle is moved by $\approx 42\%$ in the vertical direction while rest of the scene is preserved. We note that the noise is generally concentrated around the targeted instance and the region to which the instance is moved.

to which the instance is moved. For example, when moving a vehicle right or left, the corresponding perturbations also move right or left.