[Reviews · NeurIPS 2020]

Review 1

Summary and Contributions: The paper presents an experimental study of robustness to adversarial attacks for monocular depth estimation networks. The attacks are image- and model-specific, obtained by constrained reconstruction of target depth maps via a small additive image noise. The optimization is done iteratively via SGD on the absolute relative depth error with clipping of the noise image to values in the 2e-3 - 2e-2 range. The experiments with stereo-trained monodepth and monodepth2 networks on KITTI show that it is possible (although gradually more difficult) to: - scale the entire scene forward or backward by 5-10% with ~1% error rates and noise l1 norm of ~[1e-2, 1e-1]; - horizontally flip the scene with ~15% relative error and noise l1 norm of ~3e-1; vertically flipping is much harder (often fails, error ~50%) due to the dataset bias (as expected due to gravity present as a natural inductive bias in the data) - fit a target depth map (e.g., without trees, walls, cars, etc), although the error rate is not clear and the noise l1 norm seems high (0.362); - scale or even remove specific categories or even instances in the predicted depth maps with attacks that are i) global (full image, ~5% error), ii) non-local (outside object masks, qualitatively shown possible as the depth networks rely heavily on contextual information to solve the ill-posed geometric inverse problem of lifting 2D pixels back to 3D), or iii) local (inside object masks, not always possible, qualitatively harder); - create multi-network attacks by training jointly or summing individual attacks, although they do not transfer from one model to another; - improve a bit the robustness via Gaussian blurs or adversarial training, although not alleviating the issues completely; - also attack indoor scenes similarly (e.g., on NYUv2).

Strengths: S1: Motivation. As monocular depth estimation is a very popular topic with clear applications in robotics, including safety-critical contexts like autononous driving, this experimental study is providing valuable insights into the robustness of depth networks. The questions experimentally investigated are particularly interesting and well-posed in terms of practical implications. The attack mechanism itself is simple (minimizing the relative l1 error via SGD and noise clipping). S2: Experimental evaluation. The evaluation is thorough, although some parts are missing quantitative results and a large portion of the study is in supplementary material (e.g., the impact of defenses, NYUv2, etc). Many attack scenarios are studied (cf. summary above). Variance is also systematically evaluated.

Weaknesses: W1: Perceptibility. It is difficult to assess whether the attacks are perceptible, and the corresponding claims are not directly supported. Is 2e-2 noise visible when the total variation is 0.362? Furthermore, the images are low resolution (to no fault of the authors, that is just a constraint of the medium), so it is hard to assess qualitatively. One potential suggestion here could be to conduct a small user study at the native resolution (A/B test: is this image attacked or not?). There are only 200 images, so that might be easy to do in a lab or on a crowdsourcing service like MTurk). Would it also be possible to study l1 or l2 constraints instead of l_inf? Would the conclusions hold and would the noise be more or less perceptible? Finally, it would be great to also see if the noise is stable in time (flickering would be very perceptible I suppose). W2: Metrics in meters. Although the models are specifically selected for their ability to make metrically scaled predictions, the metrics are not reported in meters. It would be great to at least bin relative errors according to absolute depth ranges (e.g., every 5 meters) to get a more grounded interpretation of the results. W3: Lack of some quantitative results. Most of the experiments are quantitatively evaluated and discussed, but some are only qualitative, for instance, fitting a target scene (4.3). In addition, it is unclear how some of the attacks are selected. For instance, in 4.3: how are target scenes built? One easy, systematic, and potentially interesting experiment here could be using the depth maps of previous or next frames in the video sequence. This would be similar to scaling but with more complex ego-motion (6dof transformation), parallax, and dynamic object motion. W4: Transferability. The experiments show that the attacks are not easily transferable (which is good in practice). This was a bit surprising because monodepth and monodepth2 are in fact quite close. It would be interesting to go beyond those (which are based on well-known attackable ResNet backbones designed for classification initially), e.g., by using the state-of-the-art and more complex 3D-convolution based PackNet model ("3D Packing for Self-Supervised Monocular Depth Estimation" Guizilini et al, CVPR'20), which has publicly released pretrained metrically-scaled models (https://github.com/TRI-ML/packnet-sfm). Another interesting class of depth estimation networks could be classification based ones like DORN. W5: Broader impact. Adversarial attacks are a complicated topic, ethically speaking. Is it better to publicly disclose attacks first (without proposing a defense mechanism, although two are studied here)? In this case, I personally think that these concerns are minor in this particular instance, as the method is not practical yet (image and model specific, 12s per image). This should be made more clear in the paper (abstract, introduction, conclusion) to avoid sounding involuntarily alarmist. Nonetheless, the potential for non-local attacks shown here is worth considering carefully (putting an adversarial billboard that makes pedestrians "disappear"). The difficulty of purely local attacks are also worth pondering (e.g., someone wanting to "disappear" from the predictions, by using "adversarial clothing", for instance would be difficult). My main immediate concern is how the media and the general public might (mis)interpret the conclusions of this paper (as has been the case for related research in the past). Could the authors discuss this further in the broader impact section and potentially highlight the impractical nature of the attacks at this stage (which does not detract from their existence and importance of course)?

Correctness: The maths are simple and well described. The experimental protocol is clear and well documented (including in supplementary material).

Clarity: The paper is very well-written. One suggestion could be to add a summary table of conclusions (a bit in the style of the summary above) to help convey the take-home message of the thorough experimental study. There are many interesting conclusions, but they are spread over multiple pages currently, making it difficult to get a global view of the results (including quantitatively).

Relation to Prior Work: The prior works are discussed sufficiently (although I am no expert in adversarial examples and there is very vast and fast-changing literature on that topic). One paper I was surprised to not see cited and discussed is: How do neural networks see depth in single images? Tom van Dijk, Guido C.H.E. de Croon, ICCV 2019. It dives into depth network biases and non-targeted "physical" attacks, with some conclusions in common with this paper. Note however that attacks here are completely different (adversarial noise).

Reproducibility: Yes

Additional Feedback: POST REBUTTAL UPDATE ------------------------------- The authors did a stellar job with their rebuttal! My main comments (perceptibility, metrics in meters, transferability) and other reviewers' concerns (e.g., novelty) have been concisely and often quantitatively answered. I believe the authors will be able to follow most recommendations in the final version, so I am happy to update my score to a solid 7. Good job!


Review 2

Summary and Contributions: The author proposed a targeted adversarial attack method for monocular depth estimation. The method is based on DAG (Dense Adversarial Generation [Xie etal. 2017]) and adapted for their specific tasks. They experiment KITTI images and show that they can achieve different targets, including scaling the scene, symmetrically flipping the scene or altering predictions to fit a preset scene.

Strengths: - The method is easy to understand and implement. - The attack is very effective. - They designed security-sensitive self-driving scenarios such as removing an instance or translate an instance (e.g. moving vehicles, bikers, pedestrians).

Weaknesses: - One concern is the novelty of the paper. Attacking on pixelwise prediction tasks has been extensively studied in prior works, and monocular depth estimation is only a special case. - Another concern is about motivation. We already realized that deep networks have vulnerabilities in various tasks, especially against adversarial examples. Researchers would naturally expect that a depth prediction network will be prone to white-box adversarial attacks. Using DAG, this method or other methods does not make much difference. This paper confirms that and does not bring any new surprising results. - The method is only compared to one baseline, DAG, which is re-purposed for the depth prediction task. But the methodology or baselines do not make too much difference anyway.

Correctness: The claims and the results are consistent. The empirical methodology is correct.

Clarity: The paper is clear and easy to understand.

Relation to Prior Work: The method can be considered as an adaptation of DAG [Xie etal., 2017]. It is not clearly discussed how this work differs from previous contributions other than [Xie et al. 2017].

Reproducibility: Yes

Additional Feedback: Update after author feedback: I appreciate the authors' response about the novelty. This convinced me to raise my score. Meanwhile, the difference between this method of DAG is still not large enough; it looks like only changing an objective function (and not a new one).


Review 3

Summary and Contributions: This paper studies the effect of adversarial perturbations for monocular depth estimation. On top of a recent metric depth prediction model (Monodepth2), the proposed adversarial attacks fool the network to predict specific targets. Global attacks include scaling, symmetric flipping¸ and altering to a preset depth map. Local attacks include category or instance conditioned scaling, removal, or translation.

Strengths: This work shows that depth prediction networks are vulnerable to targeted adversarial perturbations. This vulnerability study may positively impact to develop a more robust depth prediction system, in relation to safety in applications that involve interaction with physical space. Some of the proposed targeted attacks reveal potential biases learned by a monocular depth prediction network, such that the Monodepth2 model is biased to predict closer structures on the bottom half of the image and farther ones on the top half. Compared to previous works that also studied adversarial attacks, this work is specialized for monocular depth prediction and explores various targeted attacks.

Weaknesses: In is unclear if all of their claims on vulnerabilities to the proposed targeted attacks can be generally accepted for most other monocular depth prediction models except the Monodepth2 model. A transferability study between heterogeneous models (Monodepth1 and Monodepth2) in Section 6 is not really designed to address this issue but rather brief and sketchy.

Correctness: They should extend their studies to a wider range of models to support that their claims discussed in the paper are in general true for most monocular depth prediction models.

Clarity: The paper is well written and clear to understand what is happening.

Relation to Prior Work: The paper describes in the paper that compared to previous works their adversarial attacks focus on proving vulnerabilities of monocular depth prediction models and also designed to fool the network to predict a specific target.

Reproducibility: Yes

Additional Feedback: The paper needs to be extended to address if the claimed vulnerabilities to proposed adversarial attacks are widely acceptable facts for most monocular depth models. Alternatively, such overclaims could be revised that the paper is about an extensive study on Monodepth2 for targeted adversarial attacks, but then the impact of the work decreases. ====================== Update after author feedback: (Rating raised to accept) I agree that the authors did not intend to claim those discussed vulnerabilities are present across all monocular depth models. They did study adversarial attacks for indoor scenes with VNL [Yin et al. 2019] in the supplementary material, showing different responses against flipping attacks compared to the outdoor case. This is interesting and I wish they point out it in the main paper, which is missing in the current version. They also studied transferability more in depth in the rebuttal, by taking another outdoor model PackNet that does not share similar backbones as in between Monodepth and Mondepth2. This discussion should also be included in the final version.


Review 4

Summary and Contributions: This paper shows that current state-of-the-art works in joint monocular training of depth and ego-motion are vulnerable to adversarial perturbations. The authors present attacking methods under different aspects (scaling, flipping, semantic manipulation) of depth prediction with various ablations.

Strengths: The paper demonstrates that the proposed method leads to interesting results on KITTI. The paper is well written with different aspect of attacking. I like the ablation studies compared to DAG.

Weaknesses: 1. [Patch-level attacking] As far as I know, there are two types of adversarial attacking, which are patch-level perturbation and global perturbation over whole image. In the real application, the former approach is more practical as described in [Ranjan et al. "Attacking optical flow." ICCV 2019]. What is the advantage of the proposed approach? I recommend to discuss the difference between them to emphasize the contributions. 2. [Goal of work] While reading the paper, a fundamental question keeps arising: what is the purpose of altering the prediction as experiments in Sec. 4 and 5? For me, this is not clearly stated in the paper. From the practical viewpoint, the proposed attacking model is not generalizable and only works for a specific scene since it is not universal (L39). Although it can be generated in frame by frame (L128), still the purpose of this perturbation is not clear. If the proposed model helps to improve the robustness of learning depth, then the statement would be more clarified. 3. [Transferability] The authors performed their model with Monodepth2, and transferred to Monodepth as their attacking model. However, this is not enough to show the transferability. Both Monodepth models have the same training frameworks (series of SfM-learner) that jointly learn depth and camera motion, which means they both already contain the uncertainty of ego-motion while training. Obviously, this uncertainty can lead to vulnerability to those kinds of perturbation. I recommend to show the transferability along different learning frameworks for single image estimation, e.g., [Yin et al., “Enforcing geometric constraints of virtual normal for depth prediction.” ICCV 2019], which uses ground truth depth for fully supervised training. 4. [Dataset] The experiments are mainly conducted in KITTI. It is important to show how their network would generalize to other datasets.

Correctness: The paper proposes a normalized objective function in Eg. (1). However, the reason why it is better than DAG is not well-demonstrated.

Clarity: Some parts are not clearly stated. In Fig. 3, is the last row output from pretrained DepthNet only with the flipped original image? Then, is the vertical flipped image trained while pretraining? I am pretty sure that horizontal flipping is augmented in the pretraining, but not sure about the vertical flipping. In addition, why does the locally saturated region (the white wall) require little perturbation? Also please check the questions in “weakness” and “correctness” part.

Relation to Prior Work: As discussed in “weaknesses” part, I would like to see the differences between patch-level attacking, which is a different type of perturbation.

Reproducibility: Yes

Additional Feedback: Currently, I am holding borderline rating of this paper. I would like to see how the author would respond to “Weaknesses part” and the opinions of other reviewers. ### UPDATE AFTER REBUTTAL ### I update my final rating leaning towards acceptance. The authors' feedback provides convincing and promising answers to my initial review, e.g., transferability, patch-level perturbation, and results from VNL dataset. However, still I have a fundamental question on this topic: why do we need to do the single frame perturbation from a practical viewpoint? Moreover, I think this paper should be additionally reviewed by the people who are expertised in the topic of adversarial perturbation, in addition to the 3D geometry field. This concerns still remain, but overall, I update my rating.

[Author Response · NeurIPS 2020]



Figure 1: **Col 1, 2:** ARE binned every 5m for scaling 5 & 10% closer (5 & 10% farther also have similar plots). **Cols 3, 4:** perturbations do not transfer for Mono2 and PackNet [Guizilini et al] (also do not transfer for Mono and PackNet).

**Novelty (R2)**. Indeed there are few papers on attacking pixelwise predictions, none of them is *targeted* in the sense of
aiming to perturb specific regions, or objects and such that the output matches a target. We are the first to study *targeted*
adversarial perturbations for monocular depth prediction (and dense regression tasks in general). Our findings from
explainability of depth networks to properties of perturbations have not been reported by previous works. Besides the
experiments in Sec. 4-6, we recommend checking Sec. 6. of Supp. Mat., where we provided additional studies on the
responses of the depth networks to perturbations.

**Transferability (R1, R4)**. We show that perturbations *do not transfer* for Monodepth and Monodepth2 (Sec. 6).
Since Monodepth and Monodepth2 share similar backbones and losses, we thought it was sufficient to demonstrate the
lack of transferability and thus did not pursue further. Cols 3, 4 in Fig. 1 confirm that perturbations do not transfer for
Monodepth2 and PackNet (nor for Monodepth and PackNet). We chose PackNet over VNL [Yin et al] since VNL is
trained on indoors and Monodepths outdoors. We are unaware of any other *targeted* work that studies transferability.

**Comparison to DAG (R2, R4)**. We compare extensively to DAG on 4 different attacks ($\pm 10\%$, $\pm 5\%$) each with 4
different $\ell_\infty$ norms (Sec. 4.1, Fig. 2, main paper). L103-115 discuss differences between the proposed method and
DAG, drawbacks of DAG, and why the proposed approach is better suited for targeted dense regression attacks.

**Perceptibility (R1)**. We apologize for the typo, $\|v(x)\|_1 = 0.362$ should be 0.0362. We cannot provide full resolution
images here, but higher resolution samples are shown in Fig. 11, 15-17 of Supp. Mat. We conducted a study with 28
people where we ask if two images (image paired with itself or perturbed image, $\xi = 0.02$) are the same. Of the 40
random pairs (20 perturbed), users identified only 1 perturbed image on average, which verifies their imperceptibility.

**L1/L2 norm and stability/perceptibility over time (R1)**. If the norm constraint is on the entire image, then pertur-
bations may concentrate on certain regions, making it hard to guarantee imperceptibility. If it is per pixel, then the
noise can concentrate on one channel, but should still be imperceptible for the same upper bounds, $\xi$. We don't expect
flickering in the image over time since the noise is imperceptible. We will explore both directions in future work.

**Metrics in meters (R1)**. In columns 1, 2 of Fig. 1, we bin the error every 5m. Most of the error occurs around 10m.

**Missing results and selecting targets (R1)**. For the experiments where we had to hand select the target scenes, e.g.
presets (Sec. 4.3), and manipulating instances (Sec. 5.2-5.4), we chose representative samples for qualitative results
due to thousands of possible presets and manipulated instances. For preset scenes, we select 4 target scenes from the
training set that are *not* similar to those in the KITTI semantic dataset (street and highway containing cars and humans)
such as open roads and cluttered vegetation (Fig. 4). Here are their ARE for $\xi = 0.02$ ordered left to right according to
Fig. 4: $0.0316 \pm 0.0148$, $0.0288 \pm 0.0144$, $0.0237 \pm 0.086$, $0.0343 \pm 0.0161$.

**Table of conclusions (R1)**. Thanks, we will add a table of experiments and associated conclusions in the revised text.

**Broader impact (R1)**. Thanks for the warning, we will revise the text to discuss these concerns.

**Missing related work (R1)**. Thanks, we will compare against [Dijk et al.] in the revised text.

**Claims (R3)**. We do not claim these vulnerabilities are present across all monocular depth models, but we do provide
in-depth study on both supervised and self-supervised SOA methods for both indoor (VNL for NYUv2) and outdoor
(Monodepth, Monodepth2 for KITTI) settings. Monodepth2 is studied extensively and we repeat the same study for
VNL and again for Monodepth along with experiments on transferability. Fig. 1 includes an extra model (PackNet).

**Patch-level attacks (R4)**. Unlike [Ranjan et al], we propose a generic *targeted* adversarial perturbations framework
which can support both patch-level and global attacks. The patch is determined by the instance (Fig 6, Columns 3, 4)
for our masked attacks (Sec 5.3). Moreover, we have an $\ell_\infty$ norm constraint so that the attacks are imperceptible.

**Datasets (R4)**. We **do** show experiments for indoor settings (NYUv2) on VNL [Yin et al.] in Sec. 5, Supp. Mat.

**Vertical flip (R4).** Yes, the network (pre-trained models provided by authors) fails for vertically flipped images even
without an adversary. It is true, they do not augment the image with vertical flip since it may degrade "performance" on
the test set. One of the purposes in Fig. 3 is to shed light on this weakness stemming from the dataset overfitting.

**White regions (R4)**. We provide possible explanations in L135-140 of Supp. Mat. We speculate that this is due to
white being on the upper support of RGB images, which gives high activations. Hence, tweaking the intensities slightly
allows perturbations to fool the network. For more details please refer to Sec. 6, Supp. Mat.

[Meta-Review · NeurIPS 2020]

All reviewers appreciate the findings presented in this paper regarding the robustness of depth estimation networks. All reviewers recommend accept after the rebuttal. AC agrees with this consensus recommendation. The authors successfully addressed the reviewers' concerns in the rebuttal. The authors should revise the paper by incorporating the clarifications and responses presented in the rebuttal, in particular, adding the additional quantitative results and the additional discussion about transferability. NOTE FROM PROGRAM CHAIRS: For the camera-ready version, please expand your broader impact statement to include a more substantive discussion of the potential negative impacts of your work.